# *Drosophila* Sexual Attractiveness in Older Males Is Mediated by Their Microbiota

**DOI:** 10.3390/microorganisms8020168

**Published:** 2020-01-24

**Authors:** Chloe Heys, Anne Lizé, Zenobia Lewis, Tom A. R. Price

**Affiliations:** 1Institute of Integrative Biology, University of Liverpool, Liverpool L69 7ZB, UK; Chloe.Heys@glasgow.ac.uk (C.H.); annelize@liverpool.ac.uk (A.L.); T.Price@liverpool.ac.uk (T.A.R.P.); 2Institute of Biodiversity, Animal Health & Comparative Medicine, University of Glasgow, Glasgow G12 8QQ, UK; 3UMR CNRS 6553, University of Rennes 1, 263 avenue du Général Leclerc, 35042 Rennes, France; 4School of Life Sciences, University of Liverpool, Liverpool L69 7ZB, UK

**Keywords:** age, *Drosophila pseudoobscura*, female choice, indirect benefits

## Abstract

Age is well known to be a basis for female preference of males. However, the mechanisms underlying age-based choices are not well understood, with several competing theories and little consensus. The idea that the microbiota can affect host mate choice is gaining traction, and in this study we examine whether the male microbiota influences female preference for older individuals in the fruit fly *Drosophila pseudoobscura*. We find that an intact microbiota is a key component of attractiveness in older males. However, we found no evidence that this decrease in older male attractiveness was simply due to impaired microbiota generally reducing male quality. Instead, we suggest that the microbiota underlies an honest signal used by females to assess male age, and that impaired microbiota disrupt this signal. This suggests that age-based preferences may break down in environments where the microbiota is impaired, for example when individuals are exposed to naturally occurring antibiotics, extreme temperatures, or in animals reared in laboratories on antibiotic supplemented diet.

## 1. Introduction

Choosing the right mate can have a major impact on a female’s fitness [1]. Where males only provide sperm to females, females often choose mates in order to gain genetic benefits for their offspring [2,3]. One key factor that can influence the value of a male as a mate, and hence his mating success, is his age, and females in many species show preferences for males of particular ages [4,5,6]. However, there are several competing theories that suggest different reasons for how and why male quality will vary with age, and hence the age preference females should show. For example, older males may be genetically superior to young males as they are proven survivors, potentially indicating that they possess fewer maladaptive alleles [7,8]. Another suggestion is that signals of quality are more reliable in older males [9]. Alternatively, older males might experience negative impacts of pleiotropic genes that enhance their success when younger but reduce their fertility and reproductive rate when older [10,11]. A build-up of harmful germ-line mutations in older males could also reduce their offspring’s fitness [12]. Currently there is no consensus on whether females should prefer older or younger males, nor how they can judge male age. The experimental data also has not reached a consensus. For example, female preference for old males has been documented in a number of species of *Drosophila* [13,14,15], with preference for both young males [16] and males of an intermediate age [4] shown in a variety of other insects. Within the Dipterans, experimental work has found female preference for young males [17,18], and old males [14], while some Coleopteran females prefer males of intermediate age [19]. Similar variation in whether older or younger males are preferred mates is also seen in the *Gryllus* genera of field crickets, for example [20]. At present, conflicting theories and a lack of empirical evidence means we have limited ability to predict when preference for older or younger males will evolve in a species. Moreover, one can question why females evolve preferences for males of certain ages. For example, a preference for older males could be a true preference, with females benefiting from mating with older males, and using some honest signal of male age to make their choice. Alternatively, older males may simply be better at harassing or manipulating females into mating, despite this not benefitting the female. 

Recently it has become increasingly clear that an individual’s microbiota can have a major impact on attractiveness. The microbiota consists of the symbiotic and commensal bacteria associated with a host that live on and within them. Although microbiota can refer to all type of micro-organisms associated with a host or a particular environment (yeast, fungus, etc.), most authors restrict its definition to bacteria [21], and we will use this definition throughout this article. Within the microbial community, a number of studies have stressed the importance of gut microbiota in particular, on the evolution of mate preferences [22,23,24,25,26]. The microbiota community associated with *Drosophila* is subject to spatio-temporal variations [27,28], and has been shown to change through development as well as ageing [29]. However, the effect of the commensal microbiota on the lifespan of *Drosophila melanogaster* has produced conflicting results, with some studies demonstrating its beneficial effects [30] and others its detrimental effects [31,32,33]. In addition, an age-related deterioration of gut homeostasis occurs during natural aging, which is affected by the presence of [34] and variation within [34,35] the *Drosophila* gut microbiota community. It is important however, to state that studies have largely focused on flies of extreme age, ranging from zero to over seventy days old [36]. Ren et al. [36] noted that the culturable bacterial load of *D. melanogaster* increased with age, but few differences were found with regards to changing bacterial diversity. As is consistent with previous results, the bacterial species present were found to be dominated by *Lactobacillus* and *Acetobacter* species. Thus, we suggest that the microbial load, rather than microbial diversity, may be particularly likely to impact on age-based behavioural preferences. However, this is currently unknown. 

In the fruit fly *Drosophila pseudoobscura* females prefer to mate with older males [14]. In this species, females are quicker to accept matings when courted by an older male, and in trials where old and young males compete for a mating, the older males typically win [14,37]. Older males also copulate for longer and potentially invest more sperm in matings with females [14], and females produce more offspring from mating with older males, although not when mating with extremely old males [37]. There are two non-exclusive potential reasons for this mating bias. First, it may be that older males are more effective at manipulating females into mating with them, being more experienced, faster, or able to exclude younger males. Alternatively, it could be that females can detect male age due to some honest signal, making this apparent preference for older males a true female choice. 

In this study we examined whether the presence/absence of microbiota underlies the preference for older males in *D. pseudoobscura*. Preference was measured in both no choice (single male) and choice (two males) competitive mating trials, where the culturable microbiota was either intact or impaired. We hypothesised that females would prefer to mate with older males, but that this preference would disappear when the microbiota was impaired. However, males with impaired microbiota might be poor at acquiring mates simply because an impaired microbiota is costly, resulting in male physical impairment. To test this possibility, we also examined whether suppression of the culturable microbiota impacted on standard measures of male *Drosophila* activity and competence.

## 2. Materials and Methods

*D. pseudoobscura* were collected in Show Low, Arizona, in 2008, with offspring from approximately 70 wild caught females combined to produce a mixed outbred population. Flies were maintained in the laboratory at a population size of 400 adults per generation for approximately 50 generations. All flies were kept and reared at 22 °C on a 12:12 h light:dark cycle. Flies were kept in standard 75 mm × 25 mm *Drosophila* vials containing 25 mL of standard *Drosophila* food composed of yeast/agar/maize/sugar [25]. Flies were moved to new vials every 4 days.

### 2.1. Manipulation of the Microbiota

The microbiota was impaired via the addition of the antibiotic streptomycin (4 mL of 10 g streptomycin/100mL ethanol solution per litre of growth medium) to the standard growth medium. Adding antibiotics to dietary medium is a common method to suppress insect microbiota [23,38], and has few side effects in *Drosophila* [39,40] when used at low concentrations. As the core composition of the *Drosophila* gut is known to be cultivable and relatively simple [41], we used culturable bacteria as a proxy to determine whether the microbiota had been impaired or not. The strain of *D. pseudoobscura* used does not carry any bacterial endosymbionts. 

In order to determine that the microbiota was in fact impaired, we analysed the *D. pseudoobscura* gut bacterial content, as performed previously [39]. Here, the whole gut of males from both the old and young treatments from both the normal and streptomycin supplemented diet, were dissected into 250 μL BHI (brain heart infusion) liquid media. The gut-solute was then transferred into a 1.5 mL Eppendorf tube and disrupted by hand using a sterile plastic pestle. From this solution, 100 μL was placed in the centre of a petri dish containing BHI agar. A sterile glass loop was then used to spread the solute across the whole plate. This was repeated three times for both ages from both the normal and streptomycin diets. The plates were then incubated at 25 °C for 72 h, after which, the plates were checked for bacterial growth and CFU (colony forming units) counts were then performed to quantify the bacterial load. 

### 2.2. Preference in No Choice Mating Trials

Recently mated females were placed on 25 mL of either standard diet (here, named Strep-) or diet containing streptomycin (Strep+) and allowed to oviposit to form two populations. At eclosion, virgin males were isolated twice daily from each diet type to form the “old” male experimental treatment (10 days old). Eight days later, further virgin males were collected to form the “young” male treatment (two days old). Virgin females were collected from a separate set of vials that did not contain streptomycin. Mating trials were staggered over several days in order to ensure a high replication rate and reliability. Isolated virgin males were left to mature on the same food medium on which they were reared (either Strep+ or Strep-). Males were kept singly to avoid any potential effects of male–male interactions [42,43]. Females were stored in groups of 10 on food without streptomycin until they were four days old. Following maturation, the mating trials were conducted. Virgin females were gently aspirated onto 15 mL of standard food media and allowed to rest overnight. Mating trials were conducted during the morning, as this is when *D. pseudoobscura* is the most active in the wild [44]. Either an “old” or “young” male was then gently aspirated into the vial and the observations begun. We recorded whether or not copulation occurred, in addition to mating latency (time elapsed between male introduction and copulation), and the duration of copulation. In this experiment, female preference is reflected by the mating latency (the time it takes a male from being placed in the vial to start mating with the female), which is a commonly used indicator of female preference in *Drosophila* [15,44,45,46,47,48,49,50]. If no mating occurred within two hours, the trial was scored as a failure to mate. 

### 2.3. Preference in Choice (Competitive Mating) Mating Trials

In order to measure female preference when males can compete for matings, we set up trials as above, but placed two males in the vial with each female, one old male and one young one. In any trial, both males were Strep-, or both males were Strep+. We did not compete Strep+ males against Strep- males. In each trial, we recorded whether or not copulation occurred, in addition to the mating latency, and the duration of copulation. Wing clipping was used in order to distinguish between the two males. This is a standard technique used in *Drosophila* research that allows the simple and accurate detection of an individual [51,52,53,54]. Two days before the mating trials, virgin males were isolated under ice anaesthesia and a small section of the distal end of one wing was cut off. All males were wing clipped, with half clipped on the left wing, half clipped on the right. This was randomised across treatment in order to remove any potential bias, although previous work has found no effect of wing clipping on mating propensity in *D. pseudoobscura* [55]. 

### 2.4. Measurement of Male Activity

Impairing or removing the microbiota raises the possibility that differences in male performance could simply be due to the result of the different treatments (microbiota present or absent), with males with impaired microbiota being physically impaired, and thus less able to court the female. To test this possibility, we ran an independent test of male speed and responsiveness, the Rapid Iterative Negative Geotaxis (RING) test [56]). The RING test examines the climbing speed of flies after being knocked to the bottom of a vial, and provides a simple, repeatable and accurate measure of activity speed, which correlates well with other measures of activity and physical ability [56]. Newly emerged virgin adult males were isolated and gently aspirated into vials containing 15 mL of either Strep+ or Strep- food, according to the diet on which they were reared, at a standard density of 10 per vial. Following a 10-day maturation period, flies were transferred to a vial containing 15 mL standard food media as before, placed in the RING apparatus and left to acclimate for 15–20 min. The apparatus was then sharply tapped three times on the counter, knocking all flies to the bottom of the vial, and a picture taken following a three-second period. The flies were then left to rest for one minute, and the steps repeated, five times in total. Subsequently, each photo was examined, and the height climbed by each fly in each photo was calculated from the height of the vial and the proportion climbed by the fly above the level of the food. Care was taken to ensure that each vial contained an identical height of food. Our measure of activity was the mean height climbed by the flies in each vial over the five trials. This generated an overall mean distance climbed for both the Strep+ and Strep- flies allowing comparisons in the overall physical condition of 10-day old virgin males, with either an intact or impaired microbiota. 

### 2.5. Data Analysis

Data for the single male trials were analysed in R2.15.0 (R Foundation for Statistical Computing, Vienna, Austria) using generalised linear models. As the latency data was not normally distributed, data were square-root transformed and analysed using a quasibinomial error structure with a logit link. In each case a maximal model was constructed, and then non-significant factors removed in a stepwise process to give the minimum adequate model. Mating success in the two male trials was analysed using binomial tests. Mating failure was analysed using chi-squared tests. 

## 3. Results

### 3.1. Preference in No Choice Mating Trials

Males with impaired microbiota had a significantly longer mating latency (impaired young males *N* = 58, impaired old males *N* = 38, intact young males *N* = 73, intact old males *N* = 61, F test: F1,153 = 6.592, *p* = 0.011) than males with intact microbiota (Figure 1). Age had no significant effect on copulation latency (F test: F1,152 = 0.022, *p* = 0.883), regardless of whether the microbiota was intact or impaired (F test: F1,151 = 0.009, *p* = 0.924).

Copulation duration was directly affected by male age (Figure 2), with older males copulating for significantly longer than young males (impaired young males *N* = 58, impaired old males *N* = 38, intact young males *N* = 73, intact old males *N* = 61, F test: F1,154 = 44.71, *p* < 0.001). Whether the microbiota was intact or impaired had no significant effect on copulation duration (F test: F1,153 = 0.181, *p* = 0.671), nor did the interaction between age and microbiota (F test: F1,152 = 0.142, *p* = 0.707).

In the no choice mating trials, young males with an intact gut microbiota failed to mate with a female significantly more than old males (young: *N* = 31 successful, *N* = 27 unsuccessful, old: *N* = 32 successful, *N* = 6 unsuccessful, X^2^ = 8.315, df = 1, *p* = 0.003). When the gut microbiota was impaired, young males similarly failed to mate with a female significantly more than old males (young: *N* = 42 successful, *N* = 31 unsuccessful, old: *N* = 51 successful, *N* = 10 unsuccessful, X^2^ = 9.446, df = 1, *p* = 0.002). No difference was observed in mating failures between old males with an impaired microbiota and old males with an intact microbiota (intact: *N* = 32 successful, *N* = 6 unsuccessful, impaired: *N* = 51 successful, *N* = 10 unsuccessful, X^2^ = 0, df = 1, *p* = 1.000).

### 3.2. Preference in Choice Competitive Mating Trials

In this experiment where a female had to choose between an old and a young male with an intact microbiota, older males gained significantly more matings than their younger counterparts (number of trials: 52, number won by old male: 38, number won by young male: 14; binomial test; *p* < 0.001). However, when the microbiota was impaired, there was no difference in the success of old and young males (number of trials: 28, number won by old male: 15, number won by young male: 13; binomial test; *p* = 0.425).

### 3.3. Rapid Iterative Negative Geotaxis (RING) Test of Male Activity

Males with impaired microbiota exhibited significantly higher upwards movement in the RING test than males with intact microbiota (impaired microbiota: *N* = 245, mean ± standard deviation (SD) = 18.3 ± 7.1 mm; intact microbiota: *N* = 279, mean ± SD = 8.9 ± 4.2 mm; t-test: t = 6.005, df = 41.281, *p* < 0.001); (Figure 3). 

### 3.4. Gut Microbiota

Plates containing the gut of flies from the normal diet had substantial colony growth (Table 1). There was a stark difference in the number of bacterial colonies present between the old and young males from the normal diet treatment. There were no bacterial colonies present on plates that contained flies reared on dietary media that was supplemented with streptomycin (Table 1). This was the case for both the old and young treatments. As is consistent with previous studies detailing the low bacterial diversity of the *Drosophila* microbiota [27], this suggests that the gut microbiota has been impaired.

## 4. Discussion

Our results confirm that female *D. pseudoobscura* prefer older males in two-male choice trials. Surprisingly, this female preference for older males disappears when the males’ microbiota is impaired. In no choice mating trials we found no significant difference in how quickly old and young males were able to begin mating with a female, contrary to previous studies. However, we found that microbiota-impaired males, whether old or young, took more time to initiate mating in these no choice mating trials. These results suggest that an impaired microbiota makes males less attractive, and that this prevents females from expressing their preference for older males. Perhaps the simplest explanation for this would be that the impaired microbiota causes males to develop poorly, making them inadequate mates with limited ability to locate and court females. However, we found that suppression of the microbiota of old males had no negative impact on a simple test of physical fitness. Indeed, males with an impaired microbiota actually scored higher in the test used. Hence it is unlikely that impaired microbiota simply reduces male ability to locate and court females. Instead, we suggest that females can detect male age in older males by an honest signal, and that this honest signal is lost when males’ microbiota is impaired. 

It is often difficult to distinguish true female choice from passive female choice driven by innate differences in males: for example, if a class of males is able to court females more intensely and is more successful in gaining matings, is the male simply overcoming female resistance, or are females choosing this class of male because they gain adaptive benefits? In some models of mate choice, the question is irrelevant, but in others it is important [57]. In the current study, we used the RING test to give us a general measure of male activity. This measure correlates well with several other standard measures of *Drosophila* vigour and activity [58]. Microbiota-impaired males performed slightly better in the RING test than normal males, suggesting that there was no unintended damage to males reared with an impaired microbiota. These results are consistent with a recent study demonstrating the gut microbiota as a modulator of locomotor behaviour in *D. melanogaster* [59]. Schretter et al. [59] found that germ-free flies displayed hyperactive locomotor behaviour, including increased walking speed and daily activity, with no damaging side effects to fly physiology. With the re-colonisation of bacteria, these hyperactive behaviours returned to normal levels. Further, in *Drosophila*, copulation duration is controlled by males, and is generally correlated with male reproductive investment [60]. In this experiment, copulation duration was not altered by the microbiota impairment of males, which adds support to the fact that microbiota-impaired males do not suffer physiological alterations that would consequently affect their sexual behaviour. In a previous study, copulation duration was found to vary according to microbiota impairment, and authors acknowledged that the use of antibiotics could have more general physiological effect on the flies [25]. However, the use of low-dose antibiotics such as streptomycin, has recently been found to be more reliable and less detrimental to fly health than other methods of disrupting the gut microbiota [40]. Antibiotic use resulted in minimal impacts on a number of life history traits, including weight, development time and egg-to-adult survival. Our results suggest that streptomycin has little or no effect on male sexual abilities. Taken together, our results suggest that the impaired microbiota may reduce older male success through disrupting some signals females use to assess potential mates. 

If females are using some potentially honest signal of male age, which is disrupted by antibiotics, what might this signal be? Perhaps the strongest candidate is the cuticular hydrocarbons (CHCs) that are a key sexual signal in *Drosophila* [61] and many other insects [62]. CHCs are widely referred to as sex pheromones, due to their communicating essential information to a potential mate. For example, Scott et al. [63] noted that slight changes in the composition of CHC profiles were shown to significantly alter mating success in *Drosophila* species. CHCs are strongly influenced by diet and environment. Ageing has been shown to alter the composition of CHC profiles in both the stingless bee [64], and mosquito [65]. Similarly, in *D. melanogaster*, it has been shown that ageing alters a variety of CHC compounds, with consistent variation amongst individuals suggesting that these changes with age are strongly regulated [66]. It is possible that a male’s CHC profile provides an honest signal of age in *D. pseudoobscura*. However, if the microbiota is impaired, CHC profiles are likely to be altered, and may no longer be used/detected as an honest signal. This idea is consistent with a previous study that found kin recognition is influenced by the gut microbiota, through the disruption of olfactory sexual signalling, or CHC profile, in *D. melanogaster* [67]. Here, males displayed greater mating investment with an unrelated female when the gut microbiota was impaired, suggesting that the gut microbiota underlies CHC composition in *D. melanogaster*. Further testing in this species will require determining the CHC profiles of old and young males, with or without antibiotic exposure.

In choice mating trials, the impact of removing the microbiota only negatively affected old males. Copulation duration was similar for old and young males who had their microbiota impaired. However, microbiota-impaired old males were no longer preferred by females for mating, compared to old males whose microbiota remained intact. In *D. melanogaster*, the presence of bacteria in young males increases their longevity, while decreasing it when present in old males [30]. Although, longevity effects of the presence/absence of bacteria have not been evaluated in *D. pseudoobscura,* one can envision that the presence/absence of bacteria in male *D. pseudoobscura* may reflect their age and potential remating probabilities. Therefore, old and young male with impaired microbiota would be perceived as of similar ages by the female, while old males with intact microbiota could be perceived as having a lower probability of remating. This would likely result in old males investing more in each copulation than a young male as they are likely to have fewer remaining opportunities to mate. Indeed, old males copulate for longer than young males regardless of their microbiota status (intact or impaired) in our experiment, and in a previous study [14]. The impact that this may have on sexual selection in wild *D. pseudoobscura* could be profound. For example, in populations in which flies feed on atypical food, or are exposed to extreme temperatures, the normal microbiota may be impaired in males. This would remove the honest signal of old age in this species and potentially allow younger males to gain increased access to females, thereby overcoming the evolution of female choice.

In conclusion, we find that *D. pseudoobscura* males reared on an antibiotic-supplemented diet have decreased attractiveness to females. This effect is particularly strong in older males, which causes females to lose their preference for them. This change in attractiveness is not simply due to microbiota-impaired males having decreased energy or movement ability, because they perform better than normal males in a simple physical test, as is consistent with previous studies. Instead, we suggest that females are using an honest signal to assess male age, and that impaired microbiota damages this signal in older males. This suggests age-based preferences may break down in environments where the microbiota is impaired by natural antibiotics, unusual diets, temperature extremes, or in animals reared in laboratories on an antibiotic-supplemented diet.

## Figures and Tables

**Figure 1 microorganisms-08-00168-f001:**
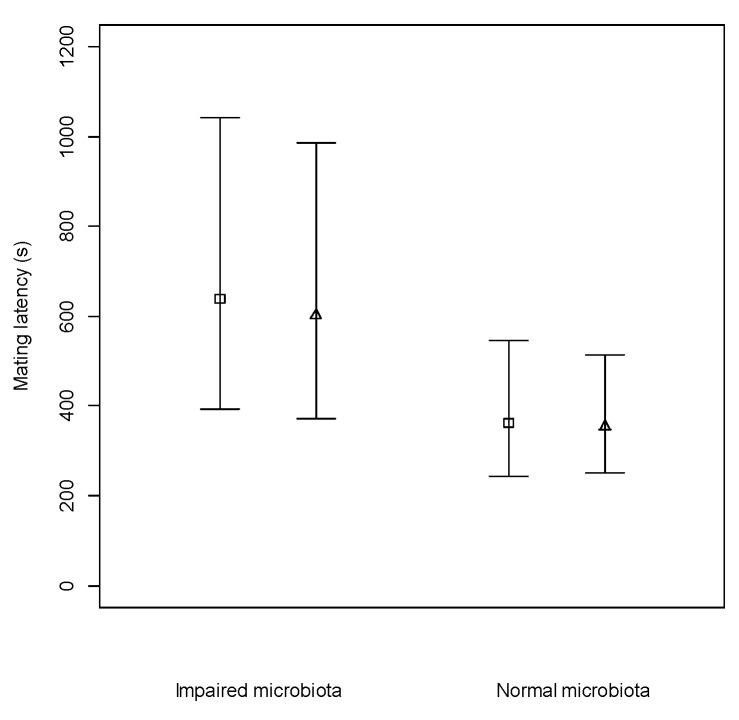
Mean copulation latency and 95% confidence intervals of old (square markers) and young (triangular markers) males, with either their microbiota impaired or intact (Normal) when placed with a single female.

**Figure 2 microorganisms-08-00168-f002:**
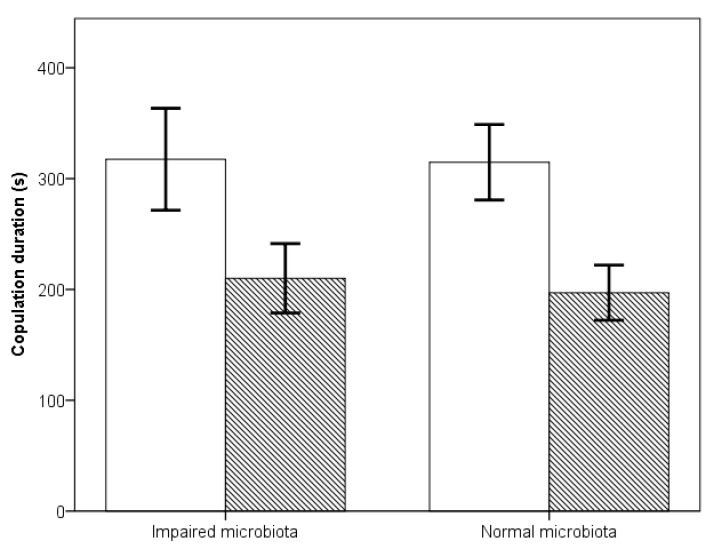
Mean copulation duration and 95% confidence intervals of old (open bars) and young (hatched bars) males, with either their microbiota impaired or intact when placed with a single female.

**Figure 3 microorganisms-08-00168-f003:**
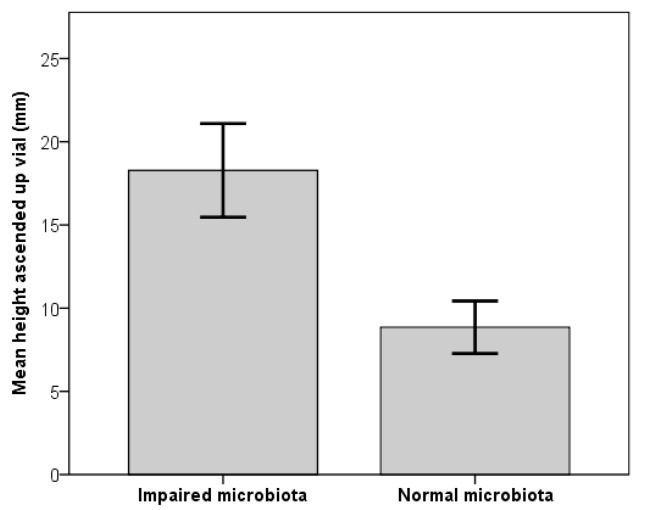
Mean height males climbed up a vial after being knocked to the base, using the Rapid Iterative Negative Geotaxis (RING) test, with either their microbiota impaired (Strep+) or intact (Strep-). Error bars show 95% confidence intervals.

**Table 1 microorganisms-08-00168-t001:** Bacterial colony counts of the whole gut from male flies of both old and young ages, from both normal and antibiotic-supplemented diets.

Diet	Age	Replicate	Number of colonies
Normal	Young	1	1
Normal	Young	2	3
Normal	Young	3	2
Antibiotic	Young	1	0
Antibiotic	Young	2	0
Antibiotic	Young	3	0
Normal	Old	1	6396
Normal	Old	2	8528
Normal	Old	3	12,428
Antibiotic	Old	1	0
Antibiotic	Old	2	0
Antibiotic	Old	3	0

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
