# Peer review of "Drosophila Sexual Attractiveness in Older Males Is Mediated by Their Microbiota"

_microorganisms, 2020, doi:10.3390/microorganisms8020168_

Round 1
Reviewer 1 Report
The paper by Heys et al. investigates the effects of gut microbiota removal through antibiotic treatment on male attractiveness. In the manuscript, the authors claim to investigate the effects of aging and gut microbiota on mate choice in the fly Drosophila pseudoobscura. The experiments do not however fully address the questions. The authors found that removal of microbiota affects male attractiveness in older males, and speculated that the microbiota composition might be an honest signal for females to assess male age. There is however no data on changes of microbiota composition with age and the experiments are quite shallow without answering this question. While the paper is really well written and has great ideas, I feel that the experiments did not go far enough to address the questions proposed in the introduction. The authors need to either reframe their introduction and restrict their questions to the experiments they really did or add more experiments to fully address their questions.
Here are some major comments:
-The authors only check for elimination of culturable bacteria. It needs to be specified in the manuscript that the antibiotic treatment used here eliminates culturable bacteria but there is no data about elimination of unculturable bacteria (PCR or sequencing).
-Why Strep+ males against Strep- males was not tested?
-Data in figure 1 clearly show non equality of variances, and cannot be analyzed using GLM with a Gaussian distribution. Have the authors used another distribution to analyze the data?
Author Response
Response to reviewer 1
Thank you for your constructive comments on our manuscript.
Reviewer 1: The paper by Heys et al. investigates the effects of gut microbiota removal through antibiotic treatment on male attractiveness. In the manuscript, the authors claim to investigate the effects of aging and gut microbiota on mate choice in the fly Drosophila pseudoobscura. The experiments do not however fully address the questions. The authors found that removal of microbiota affects male attractiveness in older males, and speculated that the microbiota composition might be an honest signal for females to assess male age. There is however no data on changes of microbiota composition with age and the experiments are quite shallow without answering this question. While the paper is really well written and has great ideas, I feel that the experiments did not go far enough to address the questions proposed in the introduction. The authors need to either reframe their introduction and restrict their questions to the experiments they really did or add more experiments to fully address their questions.
Response: We agree that further data assessing how the bacterial species diversity may change according to male age would be an interesting venture. We have altered our manuscript accordingly to state that we focused on the presence/absence of microbiota found within male flies as a means of honest signalling to females. We also note that as the bacterial load differs dramatically between young and old flies (as is consistent with previous studies), that it could in fact be bacterial load, rather than a certain bacterial species attributed to differing diversities, that is responsible for this female preference.
Here are some major comments:
-The authors only check for elimination of culturable bacteria. It needs to be specified in the manuscript that the antibiotic treatment used here eliminates culturable bacteria but there is no data about elimination of unculturable bacteria (PCR or sequencing).
Response: We have added throughout the manuscript that we are referring to the culturable bacteria found within Drosophila pseudoobscura. We have also added a sentence that states the reason for this:
“As the core composition of the Drosophila gut is known to be cultivable and relatively simple [e.g. 67], we used culturable bacteria as a proxy to determine whether the microbiota had been impaired or not.”
-Why Strep+ males against Strep- males was not tested?
Response: In the current study we find that the impairment of the microbiota causes a slight behavioural change in the fly, and we therefore felt it an unfair test to place Strep+ and Strep- males into the same mating arena. We believe that testing a female with either a Strep+ or Strep- male only, removed the likelihood of introducing confounding factors into the study. For example, if we tested female preference in a choice test with both a Strep+ and Strep- male, the increased locomotion displayed by the Strep+ (microbiota impaired) male could potentially lead to mating success as the male may be more actively chasing the female to pursue copulation, which does not reflect true female choice. This idea is consistent with other studies that show the removal or impairment of the microbiota has varying side effects on the host, with streptomycin addition found to be least detrimental.
-Data in figure 1 clearly show non equality of variances, and cannot be analyzed using GLM with a Gaussian distribution. Have the authors used another distribution to analyze the data?
Response: Indeed, we used square-root transformed data analysed with a quasi-binomial error structure and a logit link function. This is mentioned in the data analysis section of the material and methods.
Reviewer 2 Report
The two-choice test nicely demonstrated that the antibiotic treatment diminished the difference between old and young males. However, I was confused with the result of no-choice test showing that the copulation latency was not different between old and young males, which is inconsistent with the previous studies.
This may affect the conclusion because the result of the two-choice test does not prove by itself that "the removal of microbiota only negatively affected old males" -- it might have positively affected young males. Of course it was clearly shown in no-choice test that the removal of microbiota negatively affected both old and young males in the copulation latency. But then it becomes more puzzling why the copulation success was not affected by the same treatment. It is also inconsistent that the intact old and young males or old intact and impaired males, which differed so much from each other in the colony counts, were equal in the no-choice test (latency and success, respectively).
Currently, the manuscript does not contain any explanation concerning this issue. I would like to know the authors' thoughts on this point and the reasons why we can consider it should not affect their conclusions.
A minor comment: Figure 1 could be a bar graph as other figures.
Author Response
Response to reviewer 2
Thank you for your constructive comments on our manuscript.
Reviewer 2: The two-choice test nicely demonstrated that the antibiotic treatment diminished the difference between old and young males. However, I was confused with the result of no-choice test showing that the copulation latency was not different between old and young males, which is inconsistent with the previous studies.
Response: This could be due to the difference in how males were stored prior to mating trials. In our study males were stored singly prior to our mating trials, whereas they were stored in groups in previous studies. Prior social conditions have been show to affect male mating behaviour (e.g. Bretman et al 2009 PRSB; Lizé et al 2012 Biol Lett). We chose to store males singly, despite this reducing the direct comparability with the previous study, because we were concerned that if one male maintained some gut flora, if kept in a group this flora could spread to the other males, reducing the effectiveness of our gut flora impairment.
-This may affect the conclusion because the result of the two-choice test does not prove by itself that "the removal of microbiota only negatively affected old males" -- it might have positively affected young males. Of course, it was clearly shown in no-choice test that the removal of microbiota negatively affected both old and young males in the copulation latency. But then it becomes more puzzling why the copulation success was not affected by the same treatment.
Response: In the no choice test, young males always showed a lower mating success compared to old males, whatever their microbiota status (impaired or intact). Hence it seems unlikely that young males could be positively affected by microbiota removal. There is no evidence to date of impairing microbiota having a positive effect on Drosophila (Heys et al 2018 Ecol Evol).
-It is also inconsistent that the intact old and young males or old intact and impaired males, which differed so much from each other in the colony counts, were equal in the no-choice test (latency and success, respectively). Currently, the manuscript does not contain any explanation concerning this issue. I would like to know the authors' thoughts on this point and the reasons why we can consider it should not affect their conclusions.
Response: As noted above, we stored our male virgins individually in order to avoid male-male interactions. Although to our knowledge social conditioning has not been explicitly tested in Drosophila pseudoobscura, as noted above, the different manner in which we stored virgin males could explain the difference between our study and previous studies. Work in Drosophila melangaster has found that males exposed to rival males prior to mating have significantly longer latencies to mate, and copulate for longer (Bretman et al 2013 Evolution). In D. pseudoobscura a similar duration effect has been found, although latency to mate has not been examined, to our knowledge (Price et al 2012 J Insect Physiol). Therefore, the differences we see compared to the previous work on age in D. pseudoobscura are likely to be due to the males being kept in isolation.
-A minor comment: Figure 1 could be a bar graph as other figures.
Response: Latency data represented in Figure 1 showed non-equality of variance. They were square-root transformed and analysed with a quasi-binomial error structure and a logit link function. Therefore, these data are not represented with a bar graph, which we think are typically used to depict equality of variance. In addition, bar graphs seem to be relatively rarely used when error bars/confidence intervals are unequal (e.g. larger above than below the bar, as in Figure 1).